# The Detection of Impurity Content in Machine-Picked Seed Cotton Based on Image Processing and Improved YOLO V4

**Chengliang Zhang \*, Tianhui Li and Wenbin Zhang**

School of Mechanical Engineering, University of Jinan, Jinan 250022, China; 201921200495@mail.ujn.edu.cn (T.L.); zhangwenbin@mail.ujn.edu.cn (W.Z.)

\* Correspondence: me_zhangcl@ujn.edu.cn

**Abstract:** The detection of cotton impurity rates can reflect the cleaning effect of cotton impurity removal equipment, which plays a vital role in improving cotton quality and economic benefits. Therefore, several studies are being carried out to improve detection accuracy. Image processing technology is increasingly used in cotton impurity detection, in which deep learning technology based on convolution neural networks has shown excellent results in image classification, segmentation, target detection, etc. However, most of these applications focus on detecting foreign fibers in lint, which is of little significance to the parameter adjustment of cotton impurity removal equipment. For this reason, our goal was to develop an impurity detection system for seed cotton. In image segmentation, we propose a multi-channel fusion segmentation algorithm to segment the machine-picked seed cotton image. We collected 1017 images of machine-picked seed cotton as a dataset to train the detection model and tested and recognized 100 groups of samples, with an average recognition rate of 94.1%. Finally, the image segmented by the multi-channel fusion algorithm is input into the improved YOLOv4 network model for classification and recognition, and the established V–W model calculates the content of all kinds of impurities. The experimental results show that the impurity content in machine-picked cotton can be obtained effectively, and the detection accuracy of the impurity rate can increase by 5.6%.

**Keywords:** machine adopt; seed cotton; neural network; impurity identification; impurity rate; image segmentation





## 1. Introduction

China is a country that has a high production of cotton. With the mechanization of cotton picking increasing year after year, the impurities mixed with cotton also increased. For example, there are natural impurities in machine-picked seed cotton, such as boll shells, stiff petals, cotton branches, cotton leaves, miscellaneous dust, etc. With the increases in impurity type and quantity, the difficulty of cleaning equipment increases, which affects the rating of cotton. At the same time, the processing technology of machine-picked cotton has gradually developed. Cotton processing technology is mainly extensive, and the intelligence degree is not high, resulting in the waste of resources and energy consumption. The economic benefits are not apparent. The working conditions of machine-picked cotton, such as the high-quality equipment and complicated machinery, make a difference in cotton cleaning. For example, suppose the rotational speed of the cotton-picking processing equipment is too low: in this case, the cleaning effect of cotton is not good, and a rotational speed which is too high will cause damage to the cotton fiber [1,2]. Therefore, real-time monitoring of the impurity content in seed cotton can provide a basis for adjusting equipment parameters of machine cotton picking and processing equipment. This is of great significance for improving the quality of cotton.

Image processing technology is widely used in the field of cotton detection. With its real-time and high-efficiency characteristics, it can replace human eye detection, which

effectively saves labor costs. Identifying impurities after image segmentation adopts classification and recognition technology to obtain the content rate of various impurities in cotton, relay more detailed information, and guide the processing of machine-picked cotton. Traditional methods mostly use multi-feature fusion methods to classify and identify cotton impurities. Due to the complex features, it is not easy to determine all the features that can be effectively distinguished, as well as their weights. With the development of deep learning, the classification and recognition of cotton impurities based on deep learning target detection technology has become possible. Lu, Yang [3] and Zhang et al. [4] detected and recognized foreign fibers in cotton based on machine vision, but failed to obtain the specific impurity rate, which could not guide the cotton processing process. Dong [5] analyzed the advantages and disadvantages of various optical imaging technologies in detecting cotton foreign matter fibers and pointed out that multiple cameras and multiple light sources would be the leading technology in the future. Shi et al. [6] used image multi-resolution differences to detect foreign fibers in cotton. Wang et al. [7] analyzed the applicability of four edge detection methods for raw cotton images and obtained a better high and low threshold for the canny operator. Ding and Zhang [8] used a canny operator to detect colored foreign fibers and segmented impurities without classification and identification. Zhao et al. [9] and Zhao et al. [10] used improved particle swarm and ant colony algorithms to select the characteristics of foreign fibers in cotton to classify and recognize them. Wang et al. [11] proposed a segmentation algorithm for heterogeneous lint fibers. Mustafic et al. [12] studied the fluorescence reflectance of foreign cotton matter under different spectra. He et al. [13] used deep learning to detect foreign fibers in cotton. Wang [11], Mustafic [12], He [13], and others all tested the foreign fibers in lint, not involved in the field of seed cotton. Zhang [14,15] applied the regional color method and genetic algorithm to optimize the support vector machine (SVM) parameter model to segment the impurities in the machine-picked cotton image, which is more complex in the process of feature selection, and the recognition rate of impurities is not high. Wang et al. [16], Wang, Li [17] and Wei et al. [18] proposed a method for identifying foreign fibers in cotton based on machine vision. In conclusion, the detection of cotton using image processing technology shows that most types of cotton are for lint, and most impurities are heterotypic fibers, whereas the detection of impurity content of machine-picked seed cotton is a relatively unexplored field. The method used to classify and identify cotton impurities is relatively traditional, the feature selection process is complicated, and the accuracy is insufficient.

In this paper, based on image processing technology and an improved YOLO V4 [19] (You Only Look Once) network model, the impurity content in machine-picked seed cotton was studied, and the impurity content in cotton processing was fed back to guide cotton processing. This study used SVM segmentation, k-means clustering, and multi-channel fusion segmentation methods to segment the impurities in the machine-picked cotton image and used the impurity pixel area ratio in the image to obtain the impurity rate. SVM segmentation means that the image is divided into a foreground image and background image by the region segmentation method, and then the feature vector of foreground image is extracted as an SVM training sample to realize the semantic classifier. The classification standard of k-means clustering is based on the similarity between things. Multi-channel fusion segmentation refers to image segmentation using the saturation channel (S channel) and component channel from blue to yellow (b channel). In addition, as a comparative experiment, the mass ratio obtained is compared with the impurity ratio obtained based on image information by manually separating impurities and weighing them with an electronic balance. Additionally, for the divided impurities, the improved YOLO V4 network model is used for classification and identification, the content of various impurities is obtained, and the content of various impurities obtained by the quality method is compared. The volume–weight (V–W) model establishes a connection between image information and quality, and the impurity rate based on quality is obtained.

## 2. Materials and Methods

In order to solve the problem of rough segmentation effects and complex recognition in the process of machine-picked seed cotton impurity detection, a multi-channel fusion segmentation method and an improved YOLO V4 classification and recognition method are proposed in this paper.

### 2.1. Construction of Image Acquisition Test-Bed for Machine-Harvested Seed Cotton

The test material was machine-harvested seed cotton, and the impurities included boll shells, cotton branches, weeds, and leaf debris. The camera chosen was a YLR-SN85 with a resolution of 3840 t by 2160 t, a cmos size of 2.7 t and a frame rate of 30 frames per second, with a USB-powered output. The maximum aperture of the lens is 1.4. The focal length is 5–50 mm, and 3 million pixels. The illumination light source is a 4-segment strip LED diffuse light source AFT-WL21244-22W, and the light source controller is AFT-ALP2430-02. The light source was installed on an adjustable bracket, which was convenient to adjust the irradiation angle of the light source. At the same time, the light source bracket was fixed on the bench profile through T-shaped nuts and bolts, which could adjust the distance between the light source and cotton to provide high brightness and balanced lighting for the system. The camera was fixed above the cotton and the light source by a cantilever bracket and was connected to the computer via a USB. When the system was working, the cotton was manually placed on the bottom of the table, and the high-definition camera completed static shooting of the image by adjusting the distance of the light source and the irradiation angle, and then the computer performed image processing. After sampling, the next batch of seed cotton to be tested was manually replaced.

A photograph of the image acquisition system is shown in Figure 1.

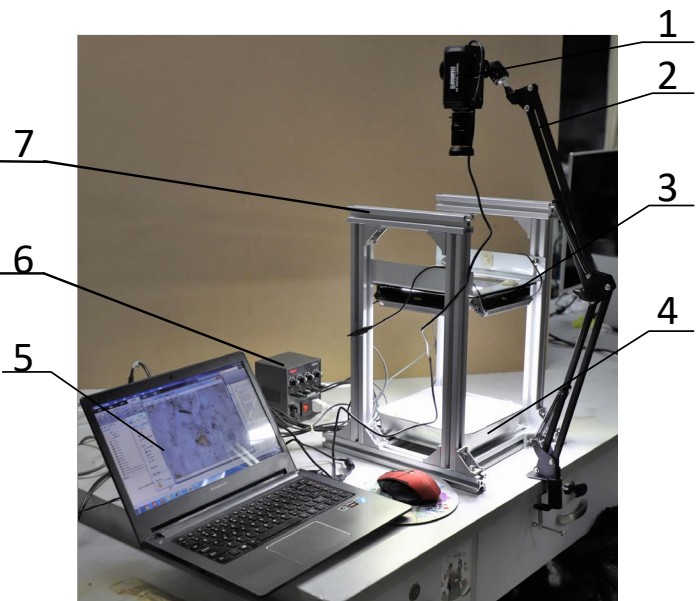

**Figure 1.** Picture of the image acquisition test bed. (**1**). Camera; (**2**). Camera holder; (**3**). Light source; (**4**). Cotton box; (**5**). Computer; (**6**). Light source controller; (**7**). Brace.

### 2.2. Impurity Segmentation of Machine-Picked Cotton Based on Image Processing

In the field of cotton impurity image segmentation, the techniques are mainly gray threshold segmentation, extracted feature segmentation, region segmentation, edge segmentation, etc. These methods have achieved a certain degree of experimental results. However, there are still limitations, such as inaccurate segmentation and the mis-segmentation of impurities with colors similar to cotton. Using a classification neural network for segmentation has a better segmentation effect, but it requires sufficient preliminary preparation and a large amount of data for training. Therefore, according to the characteristics of

machine-harvested seed cotton, this paper proposes a multi-channel fusion and segmentation algorithm.

### 2.2.1. Multi-Channel Fusion Segmentation Algorithm Flow

Impurity distributions in machine-harvested seed cotton is uneven, and the color is different. The light-colored impurities in the seed cotton are similar in color to the seed cotton. It is easy to mistakenly classify the light-colored impurities as seed cotton during the segmentation process, affecting the subsequent operations. Therefore, this article treats light-colored impurities and dark-colored impurities separately. The saturation of dark impurities in the image is higher than that of the cotton background. The dark impurities can be segmented by converting to the HSV (hue, saturation, value) color space and processing the S channel. The light color impurities have a good distinguishing effect in the b channel in the Lab color space. Through the combination of the two-color spaces, the impurities are fully segmented. The specific algorithm flow is shown in Figure 2.

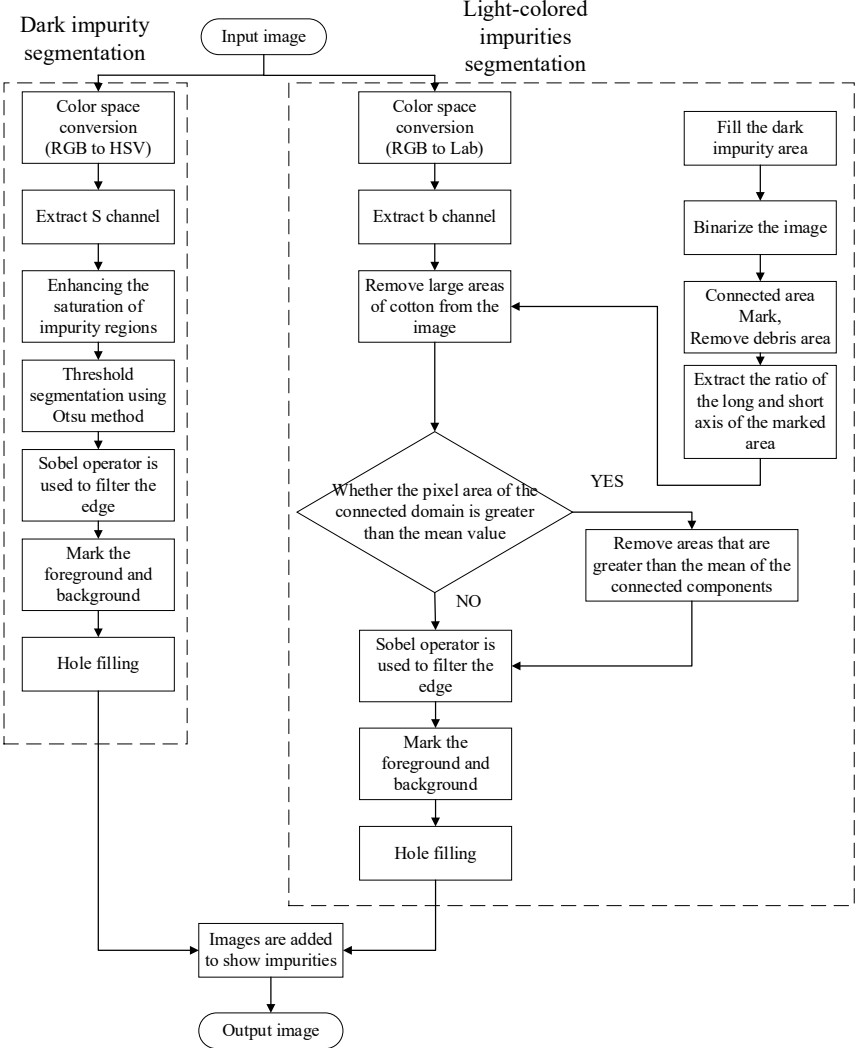

**Figure 2.** Multi-channel fusion segmentation process of impurity images of machine-picked seed cotton.

### 2.2.2. Dark Impurity Segmentation

The saturation of dark impurities in the image is higher than that of cotton background, so the color image of machine-picked seed cotton, as shown in Figure 3a, is converted to the HSV color space, and S-channel extraction is performed, as shown in Figure 3b. The

image extracted by the S channel is enhanced by the expansion and erosion algorithm in morphology to enhance the contrast of impurities and background, which is convenient for subsequent segmentation processing, as shown in Figure 3c, using Otsu [20] (an algorithm for determining the threshold of binary image segmentation) to threshold the image after image enhancement, as shown in Figure 3d; the image after threshold segmentation is filtered by the Sobel operator [21] (a 3 × 3 isotropic gradient operator for image processing), and the edges in the *X*-axis direction and the *Y*-axis direction are determined. After determining the edge of the image, the foreground and background in the marked image are reconstructed by morphology, and the holes are filled to obtain a dark impurity image, as shown in Figure 3e.

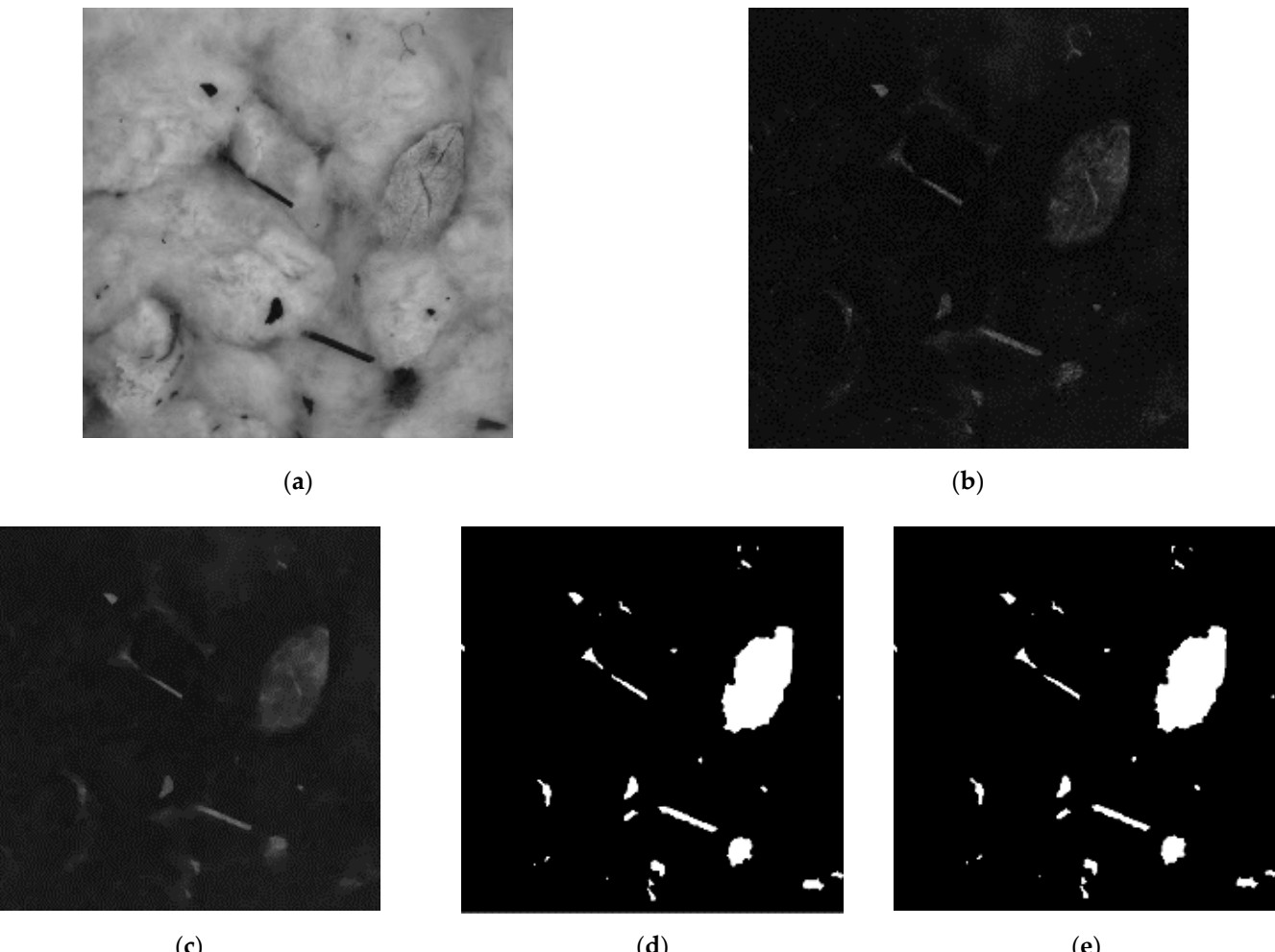

**Figure 3.** Segmentation of the dark impurity image under S channel. (**a**) The original image; (**b**) extraction of the S channel; (**c**) image enhancement; (**d**) threshold segmentation; (**e**) hole filling.

### 2.2.3. Light-Colored Impurities Segmentation

Light color impurities have a good distinguishing effect in the b channel of the Lab color space; thus, the color image of machine-picked seed cotton, as shown in Figure 4a, was converted to the Lab color space and extracted in the b channel, as shown in Figure 4b. The opening and closing operations in morphology were used to fill the dark area for the b channel, and perform binarization processing on the filled image, and mark the connected area, as shown in Figure 4c; the pixel area feature of the connected area and the long axis and short axis features of the same standard second-order central moment ellipse with the connected area were extracted, and the small cluttered area through the pixel area was removed, as shown in Figure 4d; the ratio, M, of the major axis to the minor

axis was calculated for the same standard second-order central moment ellipse of each connected region and its mean value $\overline{M}$. Comparing M with $\overline{M}$, if M was greater than $\overline{M}$, the connected area was judged to be a cotton background, and the pixel value of the area was set to 0. If M was less than $\overline{M}$, the connected area was judged as an impurity area, and the pixel value of this region was set to 1. This method was used to remove large areas of cotton background, as shown in Figure 4e. Among them, the calculation formula for the mean value $\overline{M}$ of the ratio of the major axis to the minor axis of the standard second-order central moment ellipse with the same standard as the connected region is:

$$\overline{M} = \frac{\sum_{i-1}^{n} \frac{a_i}{b_i}}{n} \tag{1}$$

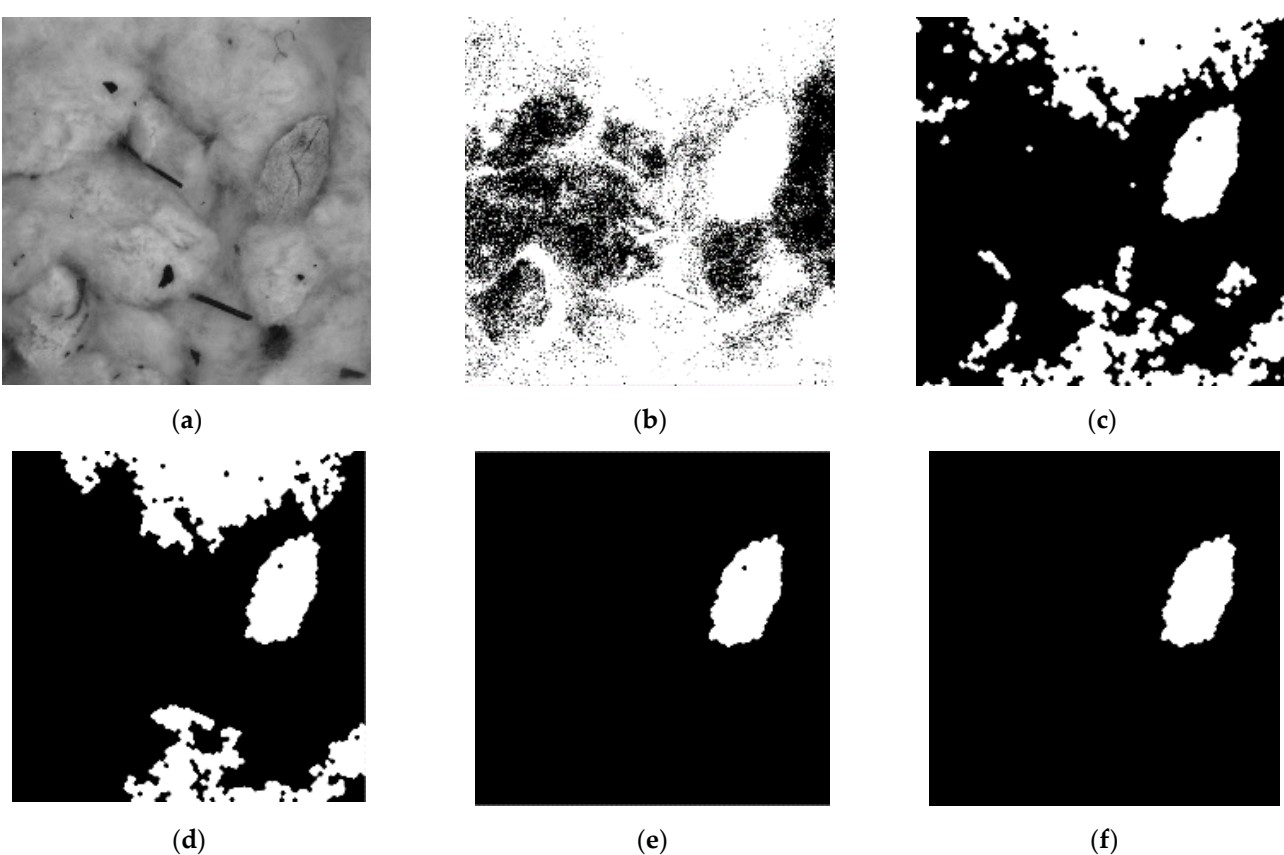

**Figure 4.** Segmentation of a light-color impurity image under the b channel. (**a**) The original image; (**b**) extracted b channel; (**c**) zone filling; (**d**) removal of small areas; (**e**) removal of the cotton background; (**f**) hole filling.

In the formula, $n$ is the number of connected regions; $a_i$ and $b_i$ are the major and minor axes of the same standard second-order central moment ellipse as the connected regions. The calculation formula is:

$$a = \sqrt{\frac{2\left(\mu_{20} + \mu_{02} + \sqrt{(\mu_{20} - \mu_{02})^2 + 4\mu_{11}^2}\right)}{\mu_{00}}} \tag{2}$$

$$b = \sqrt{\frac{2\left(\mu_{20} + \mu_{02} - \sqrt{(\mu_{20} - \mu_{02})^2 + 4\mu_{11}^2}\right)}{\mu_{00}}} \tag{3}$$

In the formula, $\mu_{pq}$ is the $(p + q)$-order mixed central moment. For the $M \times N$ image 4$f$ $(x,y)$, the $(p + q)$-order mixed central moment is calculated as:

$$\mu_{pq} = \sum_{x=1}^{M} \sum_{y=1}^{N} (x - x_0)^p (y - y_0)^q f(x,y) \qquad (4)$$

The image obtained in the previous step is filtered by the Sobel operator and the edges $G_X$ and $G_Y$ in the $X$ and $Y$ directions are obtained. The morphology is used to reconstruct the foreground and background in the marked image, and holes that may appear in the impurity area are filled to obtain a light-colored impurity image, as shown in Figure 4f.

### 2.2.4. Combine Dark and Light Impurities

A matrix addition operation was performed on the obtained binary images of dark impurities and light impurities; a binary image of all impurities was obtained, and the position of the corresponding point was found in the original image according to the position of the point with a value of 1 in the binary image; the R,G,B values of the corresponding points in the original image were assigned to the corresponding points in the image to obtain an image corresponding to the impurity position in the original image, as shown in Figure 5b.

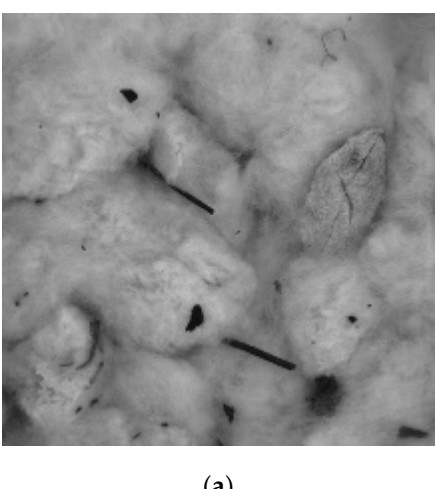
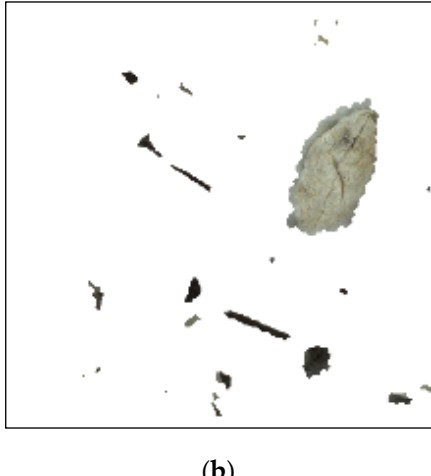

(a)          (b)

**Figure 5.** Segmentation results of impurity in machine-picked seed cotton. (**a**) The original image; (**b**) impurities after segmentation.

### 2.3. Tested Experiment on the Impurity Content of Mechanically Harvested Seed Cotton

#### 2.3.1. Impurity Detection Based on Image Information

After the machine-picked seed cotton images were collected, SVM image segmentation, k-means clustering segmentation and multi-channel fusion segmentation were used, respectively. Using the ratio of the total pixel area of impurities to the total pixel area of the entire image, the final impurity rate is obtained. The specific calculation formula is defined as follows:

$$P_s = \frac{s}{S} \qquad (5)$$

In the formula, $P_s$ is the impurity rate based on area, $s$ is the total pixel area of the impurity area in the image, and $S$ is the total pixel area of the whole image. Figure 6 is the original image of machine-picked seed cotton. The impurities include boll shell, cotton branch, weed, leaf debris, dust, etc., among which the light color impurities are similar to the background of seed cotton, which increases the difficulty of segmentation.

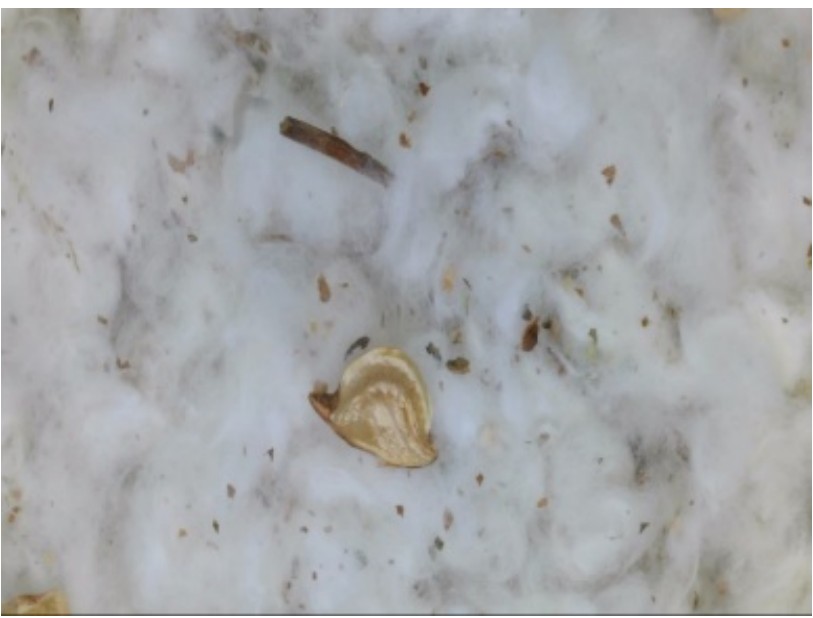

**Figure 6.** Original image of machine-picked seed cotton.

The image segmentation effect of the SVM image segmentation method, k-means clustering segmentation method and multi-channel fusion segmentation method is shown in Figure 7.

According to the segmentation results of the three methods, it can be concluded that multi-channel fusion segmentation, SVM segmentation and k-means clustering segmentation can all segment most of the impurities in machine-picked seed cotton. In the detection process, the algorithm's speed, accuracy, and robustness are the keys to judging whether an algorithm is suitable for the impurity segmentation of machine-picked seed cotton. Therefore, it is used to analyze whether the algorithm is suitable for the impurity segmentation of machine-picked seed cotton by collecting and processing several images of it.

### 2.3.2. Impurity Content Detection Based on Mass Method

According to the national standard, the quality method is the main technique to detect the impurity content. The so-called quality method refers to using the percentage of the impurity mass in the sample to the total mass of the impurity-containing seed cotton as the impurity content rate. The specific calculation formula is as follows:

$$P_m = \frac{m}{M} \tag{6}$$

In the formula, $P_m$ is the impurity content, $m$ is the impurity quality, and $M$ is the total weight of seed cotton containing impurities. In order to verify the error size of the impurity rate of machine-harvested seed cotton based on image information, the impurity rate of machine-harvested cotton obtained by the quality method was used as the comparison standard. The specific operation involves separating the impurity from the seed cotton after manual sorting of the cotton samples collected from the image, and then weighing the quality of the seed cotton and the impurity with an electronic balance with a precision of 0.001 g. By calculating the mass ratio, the impurity content of the sample was obtained.

### 2.4. Impurity Recognition of Machine-Picked Cotton Based on YOLO Neural Network

For the classification and recognition of impurities in cotton, early studies were mostly based on manual selection of features, combined with a machine learning classifier model for training, and the trained classifier was used to classify impurities according to features. On the one hand, the selected features are one-sided; thus, it is difficult to find all the features that can distinguish each kind of impurities. Therefore, the selection of features

directly affect the classification effect of the model. On the other hand, the characteristics of early machine learning model learning are relatively shallow, and the expression ability of the model is not sufficient. In addition, the improvements in its classification performance are less affected by the number of datasets.

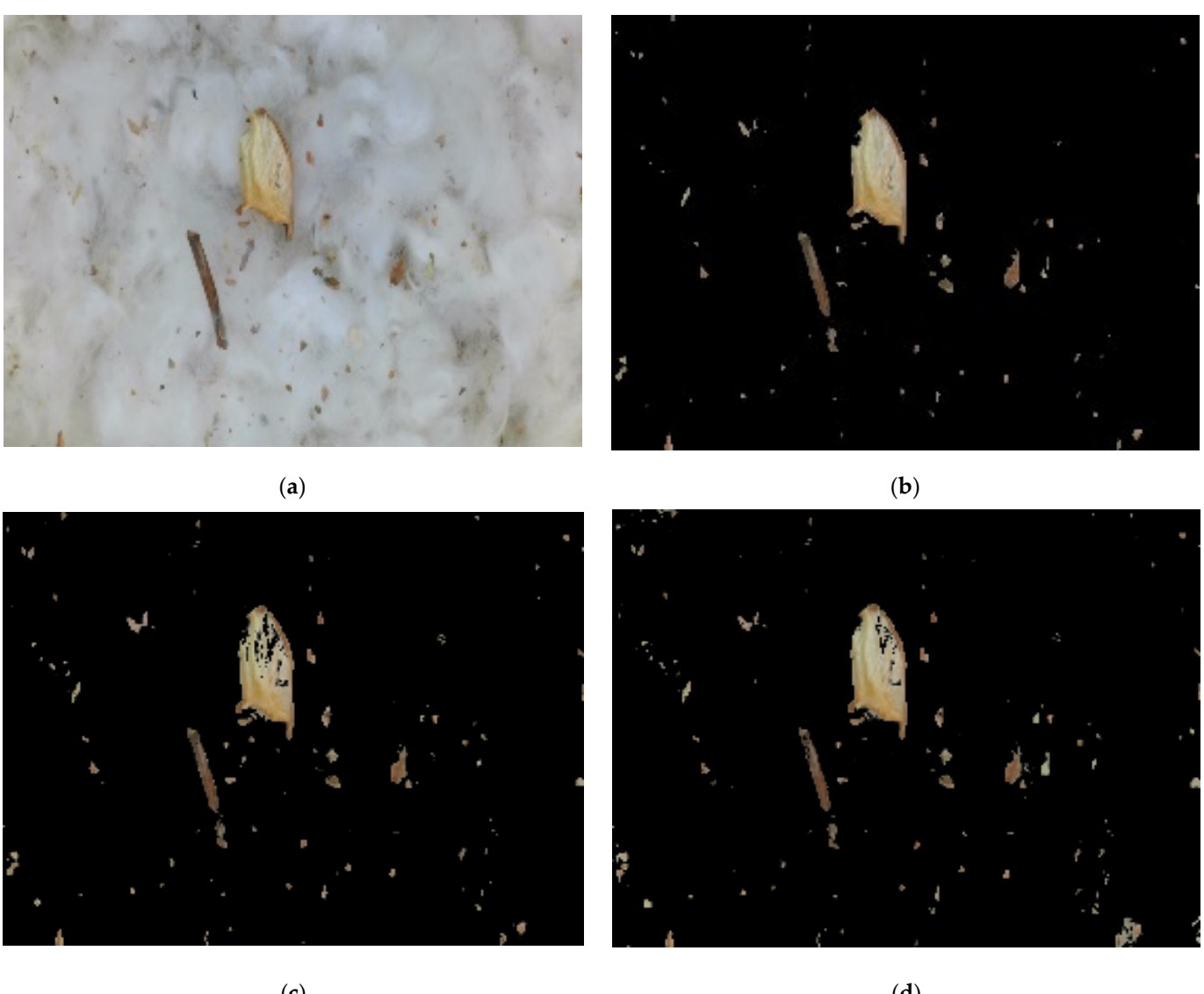

**Figure 7.** Comparison of the three segmentation methods. (**a**) The original image; (**b**) multi-channel fusion segmentation; (**c**) SVM segmentation; (**d**) k-means clustering segmentation.

A deep convolutional neural network can automatically extract and learn more essential features in the image from the training data. Applying a deep convolutional neural network to the classification and recognition technology of machine-picked seed cotton impurities will significantly improve the classification effect and locate impurities. Among them, YOLO v4 [19] integrates the advantages of FasterR-CNN, SSD and ResNet models. It is the most balanced target detection network with the most balanced speed and accuracy so far. Applications of it to the classification and recognition of machine-picked seed cotton impurities will have significant results.

### 2.4.1. YOLO V4 Model

Target detection algorithms based on deep learning can be divided into two categories: one is a two-stage detection algorithm (R-CNN); the other is a one-stage detection algorithm

(YOLO series). The former has the advantage of higher detection accuracy, whereas the latter has the advantage of higher detection speed. The core idea of YOLO is to treat the target detection as a single regression problem, and directly return the border coordinates and the category in the output layer. The input images were unified into 448 px × 448 px and then sent to the CNN for feature extraction and target prediction. In addition, sliding window technology was used in the previous target detection, which transformed the problem of image target detection into the problem of image classification. The shortcoming is that the window size is inconsistent, the number is large, and the computation requirements are huge. YOLO divides the input image into S × S grids, where each grid is used to predict the object whose center falls into the grid. This method dramatically reduces the calculation requirements of the model and further improves the detection speed of the model.

Compared with previous versions of YOLO, YOLO V4 has many updated and improved methods. Its backbone network is CSP Darknet53 [22]. In addition, a spatial pyramid pool (SPP) block of a deep convolutional network was added on CSP Darknet53 [23], which contains a total of 72 convolutional layers and a complete connection layer, using a 3 × 3 convolutional kernel. PAN [24] is selected for different levels of detectors as the method of parameter aggregation for different backbone layers, and the addition of PAN shortcut connections is changed to series. The optimization method also includes mosaic data enhancement and self-confrontation training, i.e., to let the neural network update the image in reverse, continue training after changing and disturbing the image, and adopt the Mish activation function and optimize the loss function to improve the performance in target detection.

In this paper, on the basis of YOLO V4, according to the characteristics of impurities in mechanically harvested seed cotton, the loss function is improved to improve the accuracy of detection.

### 2.4.2. Improved YOLO V4 Neural Network Loss Function

All algorithms in machine learning rely on minimizing or maximizing a certain function, known as the "objective function". The minimized set of functions is called a "loss function". The loss function can measure the prediction effect of the prediction model. When the loss function value is large, the distribution difference between the prediction model and the actual sample data is significant. Therefore, the prediction model needs to be adjusted so that the prediction model can better learn the law of the data.

In order to make the prediction model more suitable for the classification and identification of impurities in machine-picked seed cotton, the loss function was improved to guide the training of the model better.

The improved loss function in this paper is shown as follows:

$$
\begin{aligned}
Loss = bbox\ loss + &confidence\ loss + class\ loss = \\
\sum_{0}^{cell\ number*B} &I^{object} \times \left(1 - CIOU_{predict}^{ground\ truth}\right) + \\
\sum_{0}^{cell\ number*B} &m \times \sum_{y=1}^{C} CB_{focal}(CE(p_0, q_0), y) + \\
\sum_{0}^{cell\ number*B} &I^{object} \times \sum_{c=1}^{C} CE(p(c), q(c))
\end{aligned}
\tag{7}
$$

In the formula: the loss function, Loss, consists of the boundary box regression loss function, the boundary box confidence loss function, and the target classification loss function. B is the number of candidate box anchor boxes generated by each grid; $I^{object}$ is responsible for determining whether the candidate box is responsible for this object; CE is the cross-entropy loss function; $CB_{focal}$ is the improved *FOCAL* loss function; $p_0$ and $q_0$ are the probability distribution of the confidence of the real boundary box and the probability distribution of predicting the confidence of the boundary box, respectively; $p(c)$ and $q(c)$ are

the probability distribution of real target classification and the probability distribution of predicted target classification, respectively; and $C$ is the number of classes.

The loss function Loss uses (1-*CIOU*) directly as the bounding box regression loss function to replace the original mean square error and loss function, as shown in the following formula:

$$bbox\ loss = \sum_{0}^{cell\ number * B} I^{object} \times \left(1 - CIoU_{predict}^{ground\ truth}\right) \tag{8}$$

Among them, *CIOU* is as follows:

$$CIoU = IoU - \frac{L^2(R_1, R_2)}{c^2} - \alpha v \tag{9}$$

$$IoU = \frac{A \cap B}{A \cup B} \tag{10}$$

where $v$ and $\alpha$ are as follows:

$$v = \frac{4}{\pi^2} \left( \arctan\frac{w^{gt}}{h^{gt}} - \arctan\frac{w}{h} \right)^2 \tag{11}$$

$$\alpha = \frac{v}{(1 - IOU) + v} \tag{12}$$

where $A$ is the prediction box, $B$ is the actual box, and $R_1$ and $R_2$ are the center of the prediction box and the center of the actual box, respectively. $L$ is the Euclidean distance between $R_1$ and $R_2$; $c$ is the diagonal distance of the minimum closure region of the prediction box and the truth box; $\alpha$ is the parameter used to balance the proportion; $v$ is the parameter used to measure the proportion consistency between the anchor box and the target box; $w^{gt}$ and $h^{gt}$ are the width and height of the real frame, respectively; and $w$ and $h$ are the width and height of the prediction box, respectively.

The loss function Loss uses the improved Focal loss function based on the cross-entropy loss as the bounding box confidence loss function.

The focal loss function is as follows:

$$FL(P_t) = -\alpha_t(1 - P_t)^\gamma ln(P_t) \tag{13}$$

where $\gamma$ is the focusing parameter and is a hyperparameter greater than 0; $\alpha_t$ is a balance parameter, which is also a hyperparameter, used to control the weight of positive and negative samples to the total loss, and balance the number of samples in multiple categories; and $P_t$ is the label prediction probability.

In the original focal loss function, humans generally specify the parameters, and it is difficult to better deal with highly imbalanced data. Thus, referring to the function $\frac{1}{E_n}$ instead of $\alpha_t$:

$$\frac{1}{E_n} = \frac{1 - \beta}{1 - \beta^n} \tag{14}$$

$$\beta = \frac{N - 1}{N} \tag{15}$$

where $N$ is the assumed volume of the whole space, $\beta$ is a hyperparameter, which depends on the size of the hypothetical sample space, and n is the number of samples of a particular type.

After class-balancing, the *FOCAL* loss function is as follows:

$$CB_{focal}\left(P_i^t, y\right) = -\frac{1 - \beta}{1 - \beta^{n_y}} \sum_{i=1}^{C} \left(1 - P_i^t\right)^\gamma ln\left(P_i^t\right) \tag{16}$$

where $P_i^t$ represents the predicted probability of the *i*th label, $n_y$ represents the number of samples of type *y*, and *C* is the number of classes.

### 2.4.3. Algorithm Flow

The algorithm flow of the machine-harvested cotton impurity classification and identification process in this paper is shown in Figure 8.

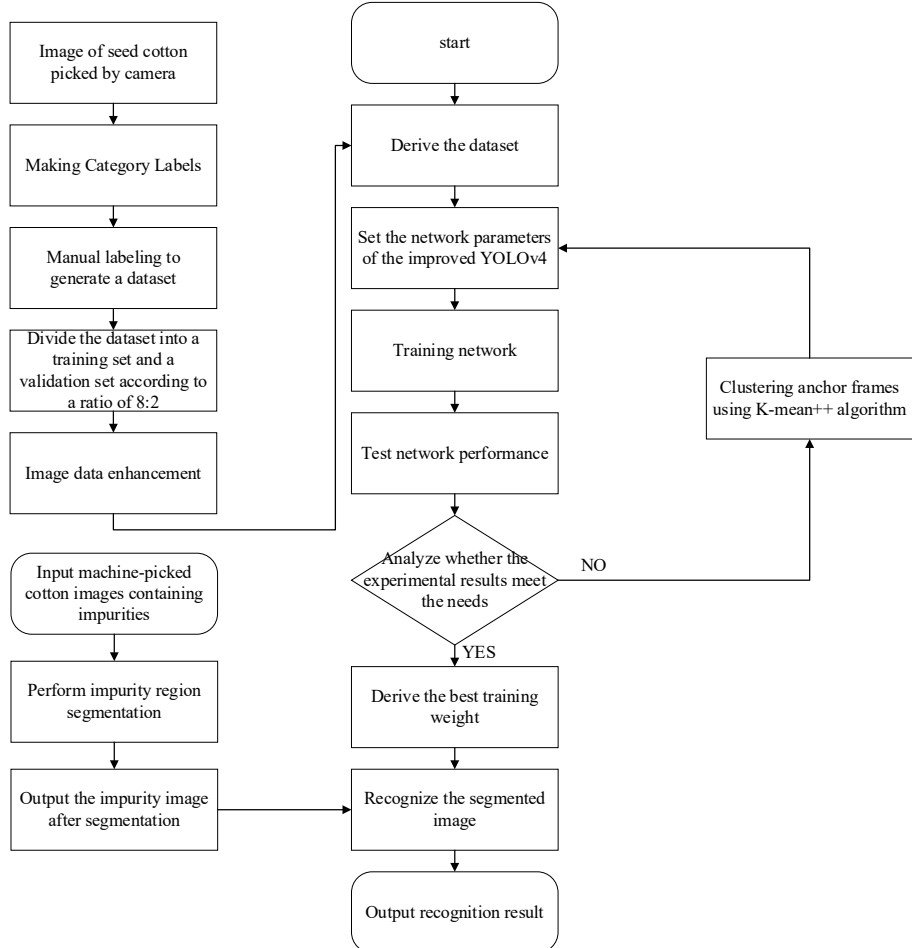

**Figure 8.** Flow chart of the impurity classification and recognition algorithm.

In this study, 1017 images of seed cotton collected by the camera were used as datasets and were divided into a training set and verification set in a ratio of 8:2. Then, the dataset was input to the YOLO v4 neural network with an improved loss function for training. The training curve is shown in Figure 9. The size of the anchor frame was continuously optimized through the k-mean++ algorithm, and a weight file was generated that meets the requirements. Additionally, the image was input after multi-channel fusion and segmentation into the trained neural network to complete the classification and recognition of impurities. Further statistics on the content of each type of impurity in the machine-collected seed cotton samples.

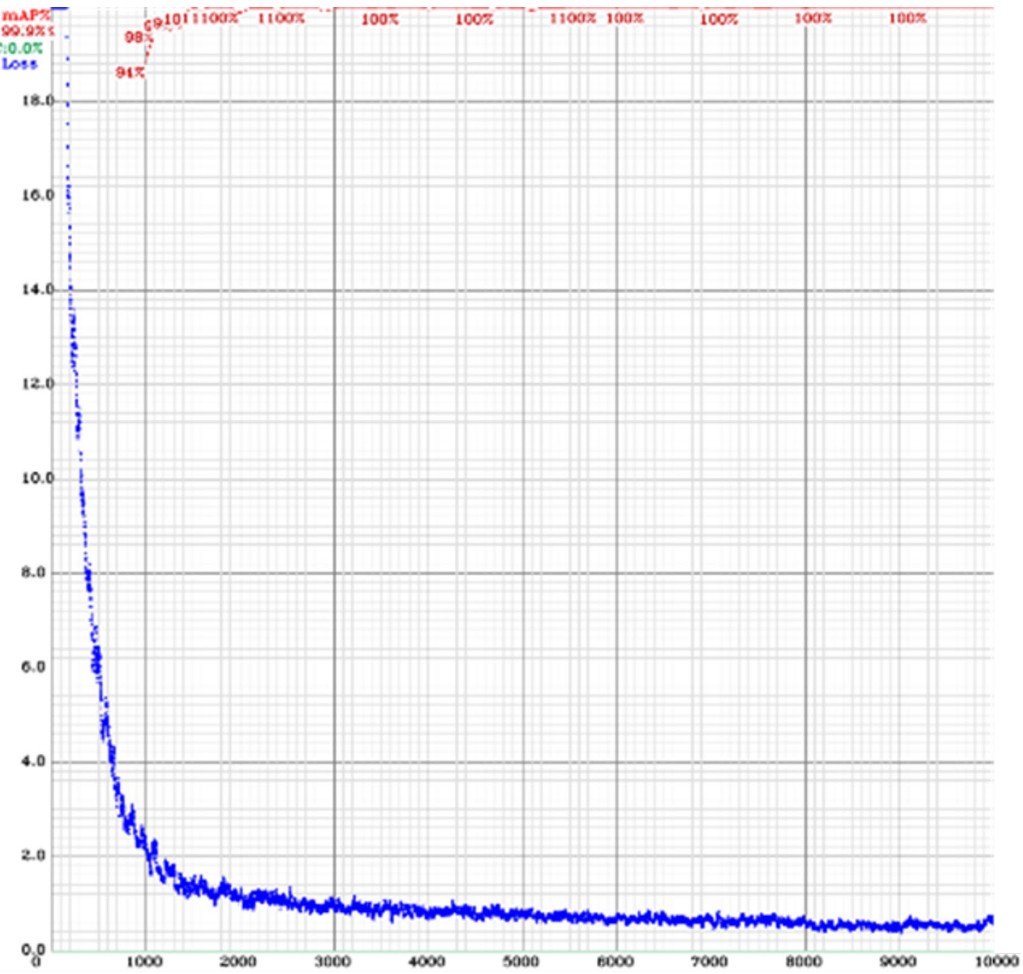

**Figure 9.** Training curve.

The machine-harvested seed cotton photos were input into the YOLO v4 network model before improvement and the YOLO v4 network model after improvement, and the classification and recognition results are shown in Figure 10.

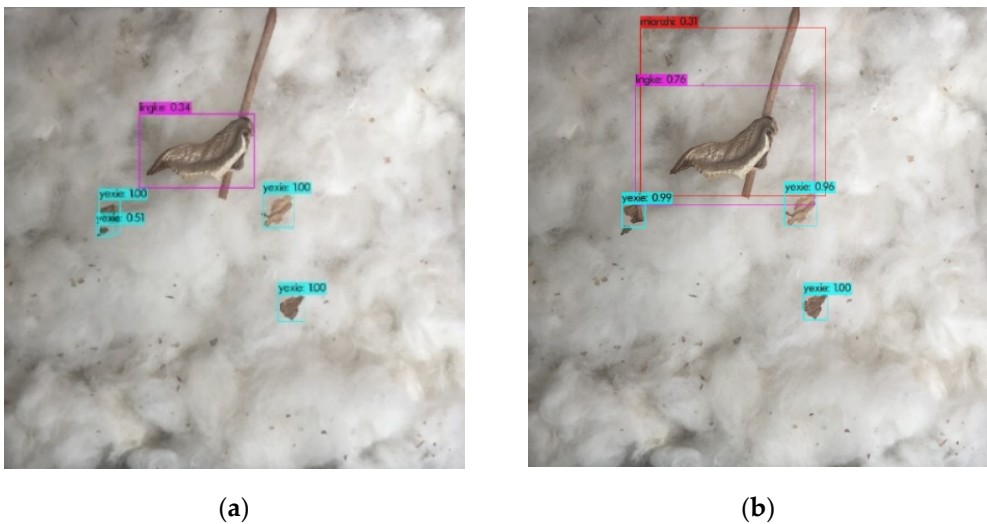

(**a**)         (**b**)

**Figure 10.** YOLO V4 network model recognition results. (**a**) Recognition result before improvement; (**b**) recognition result after improvement.

It can be seen from Figure 11 that the improved YOLO v4 network model can effectively identify the impurities that obscure each other, and the recognition effect is better.

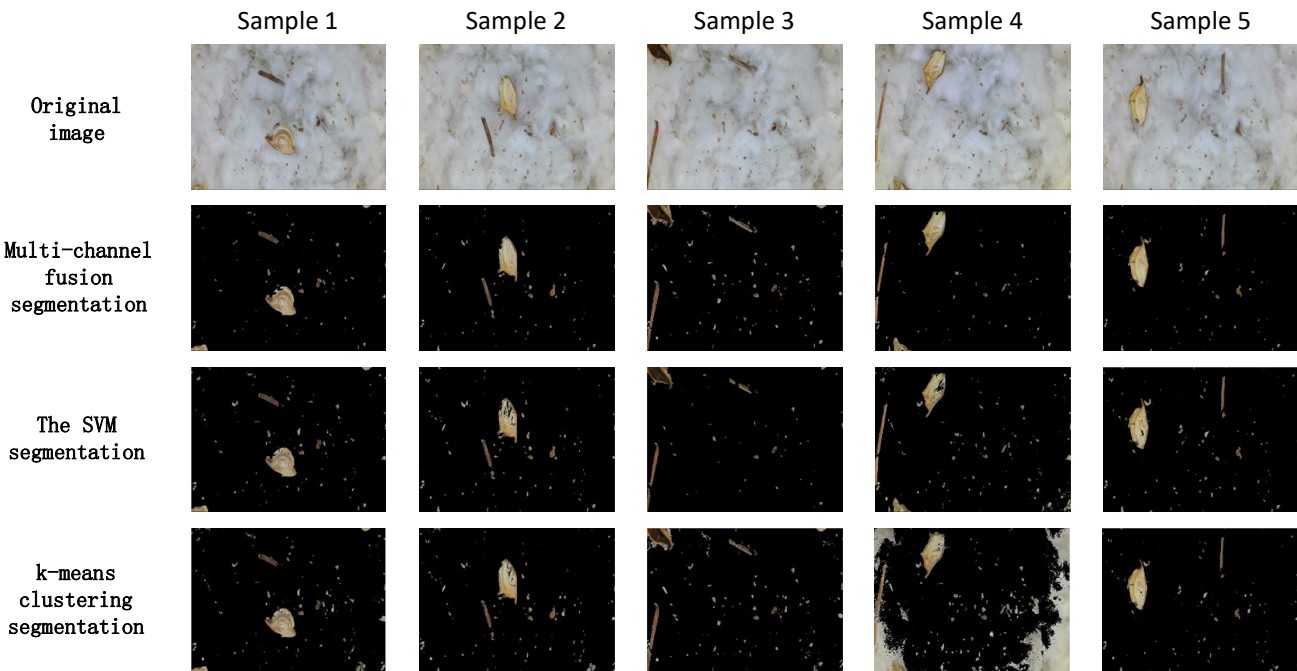

**Figure 11.** Segmentation effect diagrams of the three segmentation methods.

## 3. Results

Figure 11 shows the processing results of three segmentation methods for five samples in the machine-harvested seed cotton image.

As can be seen from Figure 11, the multi-channel fusion segmentation algorithm is relatively complete for impurities. Among them, the segmentation of large-area impurities (bell shell, cotton branch, etc.) is complete, but some small-area impurities (such as leaf chips) are easily omitted. In the results segmented by the SVM algorithm, the large-area impurities can easily develop holes, and the small area impurities are more comprehensive. In the results of segmentation using the k-means clustering algorithm, the segmentation effect for large areas of impurities is between the first two, and there are still some holes. The segmentation of small areas of impurities is more comprehensive, but it is easily affected by the cotton background. In the case of unevenness, it is easy to mistake parts of the cotton background as impurities for segmentation.

Using the ratio of the total pixel area of impurities to the total pixel area of the entire image, the resulting impurity rates are shown in Table 1.

**Table 1.** Impurity contents of machine-picked seed cotton based on image information.

| Sample Number | Multi-Channel Fusion Segmentation Impurity Rate/% | SVM Segmentation Impurity Rate/% | k-Means Clustering Impurity Rate/% |
|---|---|---|---|
| 1 | 3.4805 | 3.5452 | 4.0999 |
| 2 | 3.3284 | 3.4814 | 3.8145 |
| 3 | 2.9637 | 2.7156 | 2.9544 |
| 4 | 3.3195 | 3.4056 | 15.5212 |
| 5 | 3.6571 | 3.6604 | 3.6012 |

The results show that the segmentation effect of multi-channel fusion segmentation, SVM segmentation and k-means clustering is similar, although k-means clustering segmentation is prone to significant deviation and poor robustness due to the influence of

the cotton background. SVM segmentation requires the manual selection of background points and target points; therefore, the segmentation effect depends on manual selection. Multi-channel fusion segmentation has good robustness, no manual intervention in program operation, and can process cotton images in batches. By comparing the mean square deviation of data obtained by various algorithms, it can be concluded that the mean square deviation of data obtained by this algorithm is 0.065, which is less than 0.139 of SVM segmentation and 28.517 of k-means clustering, so the data obtained by this algorithm are more stable. In conclusion, the multi-channel fusion segmentation algorithm is more suitable for the segmentation of seed cotton images.

Ten samples were collected, and three segmentation methods were used to obtain the impurity content under this method and the impurity content obtained by the mass method for comparison. The specific experimental results are shown in Table 2.

**Table 2.** Impurity contents of machine-picked seed cotton based on image information and the mass method.

| Sample Number | Multi-Channel Fusion Segmentation Impurity Rate/% | SVM Segmentation Impurity Rate/% | k-Means Clustering Impurity Rate/% | Impurities Content in Mass Method/% |
|---|---|---|---|---|
| 1 | 5.8641 | 5.9351 | 7.1346 | 6.8696 |
| 2 | 3.9221 | 4.1156 | 4.2467 | 5.8432 |
| 3 | 6.8121 | 6.7168 | 7.5513 | 7.3546 |
| 4 | 4.2154 | 4.3597 | 13.5493 | 7.6421 |
| 5 | 4.3568 | 4.5461 | 4.6148 | 5.2894 |
| 6 | 5.7754 | 5.8157 | 5.8864 | 5.9943 |
| 7 | 2.9642 | 3.0671 | 3.1013 | 6.2113 |
| 8 | 4.5741 | 4.6821 | 4.6675 | 6.3838 |
| 9 | 5.8627 | 5.9211 | 5.9462 | 7.2131 |
| 10 | 4.3567 | 4.5712 | 11.5467 | 5.6243 |

The experimental results in Table 2 show that the impurity content detection results of three kinds of machine-harvested seed cotton based on image information are close, and there is an error between them and that of machine-picked seed cotton based on the quality method. In order to reduce the deviation of impurity content of mechanically harvested seed cotton based on image information and quality method, the impurity content of each kind of impurities was detected, and the impurity content of each kind of impurities in the image of mechanically harvested seed cotton was counted by classification and recognition technology.

The image segmented by the multi-channel fusion algorithm was input into the trained and improved YOLO v4 neural network model for the classification and recognition of impurities, and the recognition results are shown in Figure 12. One hundred pictures of machine-picked seed cotton were collected for classification and recognition tests, and the correct recognition rate was calculated. The results are shown in Table 3.

It can be seen from Figure 12 that the improved YOLO neural network can effectively identify the types of impurities in machine-picked cotton and indicate their positions in the figure. By calculating the ratio of the pixel area of each type of impurity to the total area of the image, the content rate of each type of impurity is obtained. The experiment was repeated several times, and the results are shown in Table 4.

According to the experimental results in Table 4, it can be concluded that the impurities in machine-picked seed cotton can be classified by classification and recognition technology to further refine the impurity contents in seed cotton, developing the understanding of the content of all kinds of impurities in seed cotton. In order to establish the relationship between image information and quality, the area impurity rate obtained based on image information is expanded into three-dimensional space, as shown in Figure 13. In the process of picking cotton with a large stacker, the value of the cotton impurity rate is relatively stable; therefore, in this paper, based on the impurity rate of the detection layer, it

was considered that the impurity rate during continuous processing was the same as the impurity rate of the detection layer, so the volume-based impurity rate was obtained.

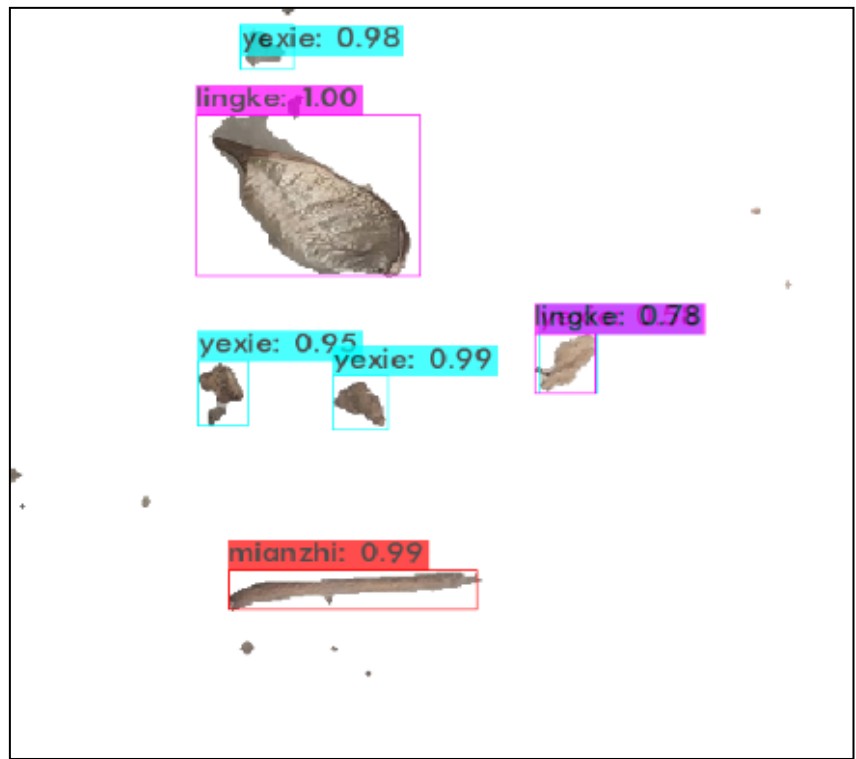

**Figure 12.** Classification and identification of impurities in machine-picked seed cotton.

**Table 3.** Classification and recognition rates of impurities in seed cotton.

| Impurities | Number of Samples | Correct Number | Number of Errors | Accuracy Rate/% |
|---|---|---|---|---|
| Leaf chip | 354 | 341 | 13 | 96.3 |
| Cotton branch | 89 | 81 | 8 | 91.0 |
| Boll shell | 76 | 68 | 8 | 89.4 |
| Weeds | 139 | 129 | 10 | 92.8 |

**Table 4.** The contents of various impurities in machine-picked seed cotton based on image information and quality method.

| | Impurity Rates Based on Image Information | | | | Impurity Rates Based on the Quality Method | | | |
|---|---|---|---|---|---|---|---|---|
| Serial Number | Boll Shell Content/% | Cotton Branch Content/% | Weed Content/% | Leaf Chip Content/% | Boll Shell Content/% | Cotton Branch Content/% | Weed Content/% | Leaf Chip Content/% |
| 1 | 2.0124 | 0.5472 | 0.2641 | 3.0453 | 1.7621 | 1.3424 | 0.2013 | 4.1635 |
| 2 | 0 | 0.4813 | 0.1430 | 3.5175 | 1.8621 | 0.6243 | 0.1380 | 4.4162 |
| 3 | 2.3441 | 0 | 0 | 2.8461 | 2.0013 | 0.2462 | 0.1121 | 3.5644 |
| 4 | 1.4382 | 1.0592 | 0.1002 | 3.1242 | 2.2490 | 0.5641 | 0.0751 | 4.1152 |
| 5 | 2.2631 | 0 | 0.1621 | 3.5317 | 1.7512 | 0.5132 | 0.1003 | 4.2822 |

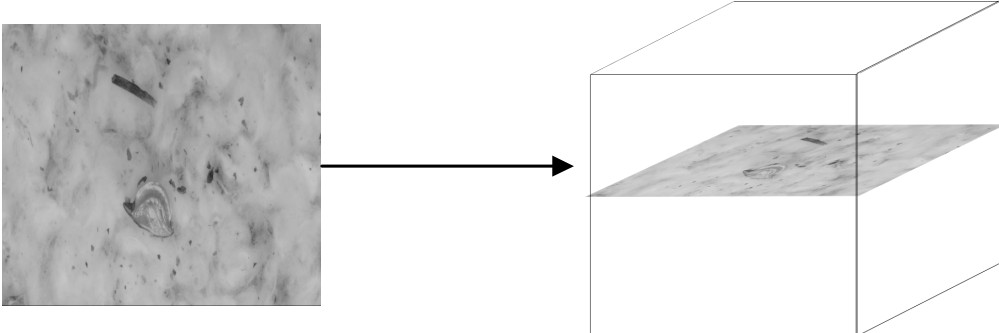

**Figure 13.** Schematic diagram of image information expansion.

The impurity area ratio obtained from the image information was used as the impurity volume ratio, as shown in the following formula:

$$P_s = \frac{s}{S} = \frac{v}{V} \tag{17}$$

In the formula, $P_s$ is the impurity content based on the area; $s$ is the area of impurities in the image; $S$ is the general area of the image; $v$ is the volume of impurities in the seed cotton; and $V$ is the volume of the seed cotton.

Moreover, by establishing a V–W model, the impurity content based on quality is calculated:

$$P_v = \sum_{i \in Z} P_{si} \frac{\rho_i}{\rho_{ZM}} = \frac{\sum M_{ZZ}}{M_{ZM}} \tag{18}$$

where $P_v$ is the impurity ratio based on the V–W model; $Z$ is the type of impurity, including bell shells, cotton branches, weeds, and leaf debris; $P_{si}$ is the impurity content of the $i$th impurity based on area; $\rho_i$ is the density of the $i$th impurity; $\rho_{ZM}$ is the average density of machine-picked cotton under the natural fluffy state; $M_{ZZ}$ is the quality of impurities; and $M_{ZM}$ is the quality of machine-picked cotton. After measurement, the density of the boll shell was 0.461 g/cm$^3$; the density of cotton branches was 0.152 g/cm$^3$; the density of weeds was 0.803 g/cm$^3$; and the density of leaf debris was 0.642 g/cm$^3$. The cotton was harvested by a machine in its natural fluffy state. The average density was 0.481 g/cm$^3$.

Using the samples in Table 4, the impurity rate calculated by the above formula and the impurity rate obtained by the other two methods are used for comparison, as shown in Table 5.

**Table 5.** Comparison of impurity contents of the three methods.

| Serial Number | Impurity Rate Based on Image Information/% | Impurity Content Based on the V–W Model/% | Impurity Content Based on Mass Method/% | Percentage Increase in Accuracy/% |
| --- | --- | --- | --- | --- |
| 1 | 5.8690 | 6.6072 | 7.4693 | 9.8 |
| 2 | 4.1418 | 5.0857 | 7.0406 | 13.4 |
| 3 | 5.1902 | 6.0454 | 5.9240 | 10.3 |
| 4 | 5.7218 | 6.0503 | 7.0034 | 4.7 |
| 5 | 5.9569 | 7.1534 | 6.6469 | 2.7 |

It can be seen from Table 5 that when all kinds of impurities are detected, the impurity content based on the V–W model is closer to that based on the quality method than that based on image information. When some impurities are not detected, the impurity content based on the V–W model is still more convergent than that based on image information. The main reason is that the distribution of impurities in seed cotton is uneven, in which the density of leaf chips is high and the content is high, which is the main factor affecting the content of impurities, the density of boll shell is small and uneven, and the influence on impurity rate is secondary. Cotton branches and weeds have little influence on the

impurity content, and image detection can effectively detect the content of leaf chips to realize the effective measurement of impurity content. We compared the impurity rate obtained based on the V–W model with that obtained based on image information, and obtained improvements in the accuracy of the impurity rate. The results show that the detection accuracy for the impurity rate improved by 5.6%.

## 4. Discussion

The proposed detection method based on image processing and improved YOLO V4 neural network model can effectively detect the impurity rate of machine-picked seed cotton collected by camera. The results show that the multi-channel fusion algorithm has good segmentation effect for both bright and dark impurities. The proposed V–W model has a smaller error in the estimation of the impurity rate of machine-picked cotton than that based on image information alone.

Compared with Shi [6] and Ding [8], Shi [6] separated different frequency information components in detection images by wavelet multi-layer decomposition for heterosexual fibers in lint, and then assessed difference between subgraphs with different resolutions to improve the contrast between the heterosexual fibers and the raw cotton information, so as to achieve segmentation. Ding [8] also targeted the heterosexual fibers in lint, using a canny operator to segment and detect. This study focused on machine-picked seed cotton, through the selection of different color channels for machine-picked seed cotton impurity segmentation. As shown in Figure 3e, dark impurities in the original figure were segmented through saturation channels; as shown in Figure 4f, the light-colored impurities in the original image were segmented through channel b. In comparison with other segmentation algorithms in Figure 11, it can be seen that in the fourth group of samples, the image using k-means clustering segmentation method has no effective segmentation around. In the same group of samples, the large impurities in the upper part of the image segmented by SVM are defective. The impurity image could be segmented well by using this algorithm. It shows that this algorithm has better robustness and more accurate segmentation performance. According to the comparison of the impurity rate obtained by several segmentation methods in Tables 1 and 2, it can be seen that the numerical value obtained by k-means clustering segmentation is unstable. For example, the impurity rate in the fourth experiment was 15.5212%, which is far from the actual value. By comparing the mean square deviation of data obtained by various algorithms, it can be concluded that the mean square deviation of data obtained by this algorithm is 0.065, which is less than the 0.139 of SVM segmentation and 28.517 of k-means clustering; therefore, the data obtained by this algorithm is more stable. Compared with the raw cotton impurity image recognition method based on edge detection [7], this study used a YOLO v4 neural network with improved loss function to train the model of impurities in a seed cotton picking machine, and input the machine seed cotton image segmented by multi-channel fusion algorithm into the model for impurity classification and recognition, which reduced the interference in the process of classification and recognition and improved the accuracy of impurity recognition. As shown in Figure 12, the image information was greatly reduced after segmentation, and most of the images were blank areas, which achieved the accurate recognition of impurities. From the data in Table 3, it can be concluded that the average recognition rate of all kinds of impurities can reach 94.1%.

It is difficult to guarantee that the model trained in this paper has good estimation performance for detecting the impurity rate of machine cotton picking in other cases. The training model was developed with machine-picked seed cotton under laboratory conditions, when it is necessary to detect the impurity content of machine-harvested seed cotton under different conditions; therefore, it will be necessary to recollect images of machine-picked seed cotton under this condition and derive a dataset for retraining. This will ensure that the retraining model is suitable for different detection environmental conditions and different machine-picked seed cotton materials.

The V–W model proposed in this paper was established by measuring the density of various impurities. By establishing the V–W model, the two-dimensional information is expanded into three-dimensional information. Under experimental conditions, the impurity rate obtained by using the V–W model is closer to the real value than that obtained solely by relying on image information, as shown in Table 5. For example, for the third group of data and the fifth group of data, although the impurity rate based on the V–W model exceeded the impurity rate based on the mass method, the error was relatively reduced. According to the data in column 5 of Table 5, the average error was reduced by 5.6%. When the conditions of seed cotton materials are different, including cotton varieties and dry and wet conditions of impurities, the density values of various impurities should be changed accordingly, so that the impurity rate of machine-picked seed cotton will have a higher estimation accuracy.

The cotton production process involves a variety of cotton processing machines, such as a cotton separator, seed cotton control box, seed cotton cleaning machine, flower machine, and lint cleaning machine; the speed adjustment of these equipment needs to be based on the rate of cotton impurities, in order to produce higher quality cotton. The impurity content obtained in this paper provides a basis for the automatic processing and production of cotton in large quantities and can guide the future processing and production process of cotton.

### 5. Conclusions

Addressing the problem of detecting the impurity rate of machine-picked cotton, this paper has proposed a method for detecting the impurity rate of machine-picked cotton based on image processing and an improved YOLO v4 neural network model. Using the segmentation algorithm of multi-channel fusion to segment the machine-picked cotton image is more robust than the traditional SVM segmentation and k-means clustering segmentation. Without manual intervention, it can effectively segment the impurities in the machine-picked cotton image.

After improving the loss function, the YOLO v4 neural network was used to train the model to recognize impurities in the machine-picked cotton. The machine-harvested seed cotton image segmented by the multi-channel fusion algorithm was input into the model for the classification and identification of impurities, and the content of each type of impurity in the seed cotton was counted.

By establishing a V–W model and measuring the density of various impurities, a connection was established between the image information and the quality of various impurities, and the impurity content based on quality could be obtained. Compared with the impurity rate obtained from image information alone, it is closer to the actual impurity rate.

**Author Contributions:** Conceptualization, C.Z.; writing—review and editing, T.L.; Validation, W.Z. All authors have read and agreed to the published version of the manuscript.

**Funding:** This work was supported by the project ZR2019MEE113 supported by Shandong Provincial Natural Science Foundation and the Key R&D Program of Shandong Province (2019JZZY020616).

**Conflicts of Interest:** The authors declare no conflict of interest.

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
