# Peer review of "The Detection of Impurity Content in Machine-Picked Seed Cotton Based on Image Processing and Improved YOLO V4"

_agronomy, doi:10.3390/agronomy12010066_

Round 1

Reviewer 1 Report

The authors improves a lot the manuscript compared to the last round, but unfortunately, they still don't provide any discussion about the results found.

My suggestion is a "major revision". Also, I encourage the authors are to improve the discussion section.

Thank you.

Author Response

Dear Reviewer:

Thank you for your comments concerning our manuscript entitled “Detection of Impurity Content in Machine-picked Seed Cotton Based on Image Processing and Improved YOLO V4”. Those comments are all valuable and very helpful for revising and improving our paper, as well as the important guiding significance to our researches. We have carefully studied the comments and have made corrections to the discussion section in line 496-561, hoping for approval. The revised part is marked in red in the paper.

Special thanks to you for your good comments.

We tried our best to improve the manuscript and made some changes in it. We appreciate for your warm work earnestly, and hope that the correction will meet with approval.

Once again, thank you very much for your comments and suggestions.

Reviewer 2 Report

The minor comments were fixed but not significant improvement was performed in the results and discussion section. Here are other minor things that need to be changed:

Line 69 -> SVM (Support Vector Machine)

Line 79 and 80 -> YOLO V4 [19] ( You Only Look Once)

Line 82 to Line 87 -> I think the organization can be improved, it looks only like copy and paste of definitions and it does not have flow with the paragraph

Line 95 -> V-W (Volume-Weight)

Table 3 -> Modify to Impurities and Acuracy Rate in table

How are the errors checked? are you manually checking? 

In table 5 the Percentage increase in accuracy/% is added but not discussed

Author Response

Dear Reviewer:

Thank you for your comments concerning our manuscript entitled “Detection of Impurity Content in Machine-picked Seed Cotton Based on Image Processing and Improved YOLO V4”. Those comments are all valuable and very helpful for revising and improving our paper, as well as the important guiding significance to our researches. We have studied comments carefully and have made correction which we hope meet with approval. Revised portion are marked in red in the paper. The main corrections in the paper and the responds to the your comments are as follow:

Responds to your comments:

  1. Response to comment:Line 69 -> SVM (Support Vector Machine);Line 79 and 80 -> YOLO V4 [19] ( You Only Look Once); Line 95 -> V-W (Volume-Weight); Modify to Impurities and Acuracy Rate in table

Response:Thank you very much for your meticulous work. We have revised the corresponding places in the paper and marked them in line 69, line 80, line 96 and Table 3 in red font.

  1. Response to comment:Line 82 to Line 87 -> I think the organization can be improved, it looks only like copy and paste of definitions and it does not have flow with the paragraph.

Response:Thank you for pointing out the problem. We have modified it and marked it in red font.

  1. Response to comment:In table 5 the Percentage increase in accuracy/% is added but not discussed.

Response:Thanks for your careful work, we have added discussion on lines 492-495 and marked it in red.

We have also revised the discussion section in line 496-561 and hope to get your approval.

Special thanks to you for your good comments.

We tried our best to improve the manuscript and made some changes in it. We appreciate for your warm work earnestly, and hope that the correction will meet with approval.

Once again, thank you very much for your comments and suggestions.

Round 2

Reviewer 1 Report

Dear Authors,

Thank you for all changes and for the opportunity to read an interesting study. Now the paper is ready to be accepted. 

Author Response

Dear Reviewer:

Thank you for your comments concerning our manuscript entitled “Detection of Impurity Content in Machine-picked Seed Cotton Based on Image Processing and Improved YOLO V4”. Wish you success in your work and a happy life.

Reviewer 2 Report

I am glad to see discussion about the limitations of the method proposed. The paper is acepted. Nevertheless, it is important to carefully check the grammar and spelling before submitting the manuscript to avoid rework.

Here are some minor changes to english and grammar:

Line 504 and 508 -> It is more professional to add the author's last name and put the number of the citation at the end of the sentence as it is shown in the introduction of this paper. Sentences should not start with numbers.

Line 508 -> Rephrase, XXX also targets....

Line 509 -> It is confusing:

This paper is aimed at machine-picked seed cotton, through the selection of different color channels for machine-picked seed cotton impurities of different shades of color segmentation

Maybe better this paper focuses ....

Line 513 to 518 -> Sentence too long, simplify

Line 526 -> stable or accurate? 

Line 544 -> established is used two times, rephrase

The V-W model established in this paper is established on the basis of measuring the density of various impurities

Author Response

Dear Reviewer:

Thank you for your comments concerning our manuscript entitled “Detection of Impurity Content in Machine-picked Seed Cotton Based on Image Processing and Improved YOLO V4”. Those comments are all valuable and very helpful for revising and improving our paper, as well as the important guiding significance to our researches. We have studied comments carefully and have made correction which we hope meet with approval. Revised portion are marked in green in the paper. The main corrections in the paper and the responds to the your comments are as follow:

Responds to your comments:

  1. Response to comment:Line 504 and 508 -> It is more professional to add the author's last name and put the number of the citation at the end of the sentence as it is shown in the introduction of this paper. Sentences should not start with numbers.

Response:Thank you very much for your meticulous work. We modified this and marked it in green on lines 503 and 507.

  1. Response to comment:Line 508 -> Rephrase, XXX also targets....;Line 513 to 518 -> Sentence too long, simplify;Line 544 -> established is used two times, rephrase;Line 509 -> It is confusing:

Response:Thank you for pointing out the problem. We have modified this in lines 507-517 and 543 and marked it in green.

  1. Response to comment:Line 526 -> stable or accurate? 

Response:Thank you for your careful work, it is indeed stable here, through the mean square error to calculate the volatility between the data, the data with high stability is more beneficial to the follow-up detection, and more accurate impurity content can be obtained.

Special thanks to you for your good comments.

We tried our best to improve the manuscript and made some changes in it. We appreciate for your warm work earnestly, and hope that the correction will meet with approval.

Once again, thank you very much for your comments and suggestions. Wish you success in your work and a happy life.

This manuscript is a resubmission of an earlier submission. The following is a list of the peer review reports and author responses from that submission.

Round 1

Reviewer 1 Report

In general, the manuscript more organization. There are terms and abbreviations not defined in the body of the text. There is not clear connection in why the different methods are being tested and more basic information about the techniques and models used are needed, thus readers not very familiar with the topic can understand the content. The most important thing that needs to be fixed is the methods and results section that are somehow mixed. There is not clear difference in between what is methodology and what is results. 

Here are my specific comments on the sections:

Abstract:

Line 11 - What SVM and K-means stand for? 

Line 15 to line 17 - In the methods section is not clear that the multi-channel fusion algorithm was used as input in the YOLO v4 network

Line 19 - The average identification rate is not presented in any table in the results section, please add these calculations for comparison in the results section

Introduction:

Line 25 - Correct word production, it is spelled product ion

Line 26 - remove capital letter in , Tthe mechanization level ...

Line 28 - change increases by increasing. However, this sentence is incomplete please check

Line 31 - missing letter the processing

Line 31 - extra letter thas

Line 31 - the processing technology has developed to what?

Line 35 - remove capital letter Ppoor

Line 36 - remove capital letter Rreal-time

Line 55 - what is a canny operator? define

Line 62 - What is SVM? spell out abbreviations the first time they appear

Line 70 - define YOLOV4 network model and add a reference

Line 73 to 75 - What are these methods? Add a simple definition for readers not familiar with these technologies and add general differences between the methods

Line 81 - What is V-M model? define

Materials and methods:

In general, it is needed to add the brand and manufacturer of the camera (line 86) and light source and its controller (line 89 and 90).

Line 112 and 113 - why is the multi-channel fusion proposed and compared with the other methods? what is the advantage? 

Line 122 - what is HSV? add meaning of HSV abbreviation

Figure 2, 3, and 4 - add in the caption that this algorithm is for the multi-channel fusion segmentation

Line 131 - what is S-channel?

Line 134 - add reference for Otsu method and definition

Line 136 - sobel operation add definition and reference

Line 146 - what is B channel ?

Line 193 - Section 2.3 --> section 2.3.1 --> This section needs to be improved. The reason of comparing the methods needs to be stated and the results of this comparison should be in the results section not here

Figure 7 - Mention the main differences between the methods here or in the introduction

From line 216 to line 247 --> this should be in results including Figure 8 and table 1

section 2.3.2 --> From line 262 to line 276 --> this should be in results including table 2

Section 2.4 --> it is not clear what method was selected for the YOLO Neural Network 

Results:

This section needs to be improved showing in a rational way the results shown in the methods section and connecting better what is learned and taken from one section to the following. In addition, as mentioned in the abstract, the identification rate of impurity classification should be also presented and discussed in this section.

Discussion:

I think in this section more information is needed about the implications of this work such as real life application, time and cost savings, etc. 

Reviewer 2 Report

Abstract

L7:  The introduction sentence is confusion. I suggest to write. Please, check out the last papers published in this journal to guide you.

Introduction:

L25: The word production need attention, and all abstract rewrite

L26-39: In my opinion, the sentence that the authors bring about the cotton mechanization level in China could be merged into one sentence. Also, they could explain better how the impurities affect the cotton picker, especially the fiber quality. I strongly suggest adding some references to endorse the affirmation about the how working condition of the machine-picked cotton harvest process can be influenced by high and slow speed rotation. In addition, the English need be improved.

L45-68: why the authors only mentioned many papers and do not comment about the limitations or differences founded in their papers compared with their own manuscript? I understood the papers mentioned the authors focused to use ML approach in cotton but not in cotton seeds, but in my opinion, it could be better written. Please improve the background presentation of this study. 

L70: This is part of M&M section…. Please provide the main aims of this study, the hypotheses.

M&M

General comments:

The table 1 and 2 could be part of result instead of m&m section. The equation 7 need be improved to better understand.

L105-114: This sentence looks like a finally introduction section that I told previously. Please consider reorganize the paper presentation.

L158: “The pixel…. Change to “t” not capitalized. Check all manuscript, I found this mistake in many parts (Capital letter in the middle of sentence).   

L301: the ten images? Please specify the number of images used in this step. Also, in line 321 let the authors know how many images were used to improved YOLO using ANN. By the way the % of data were divided into.

Results:

Figure 12: I’m not sure if the classification and recognized box is written in english. As well as at figure 11.

L444-Subscribe the number 3.

Discussion

Need be improved.